# Silicon Photomultiplier—A High Dynamic Range, High Sensitivity Sensor for Bio-Photonics Applications

**DOI:** 10.3390/bios12100793

**Published:** 2022-09-26

**Authors:** Rachel Georgel, Konstantin Grygoryev, Simon Sorensen, Huihui Lu, Stefan Andersson-Engels, Ray Burke, Daniel O’Hare

**Affiliations:** 1Microelectronic Circuits Centre Ireland (MCCI), Tyndall National Institute, T12 R5CP Cork, Ireland; 2Biophotonics@Tyndall, IPIC, Tyndall National Institute, T12 R5CP Cork, Ireland; 3Department of Physics, University College Cork, T12 K8AF Cork, Ireland

**Keywords:** auto-fluorescence, bio-photonics, diffuse reflectance, sensor interface, silicon photomultiplier

## Abstract

This work is an overview of silicon photomultipliers (SiPMs) with a view to defining their importance for bio-photonic and clinical applications. SiPMs are benchmarked against other common photodetectors, namely, PIN diodes and avalanche photodetectors (APDs) and are compared with respect to important circuit design parameters. It will be shown that careful selection of the design bias voltage, overvoltage, gain defining components and device integration to micro-optics can allow SiPM detectors to achieve considerable sensitivity for auto-fluorescence (AF) detection and a wide dynamic range at low optical powers (~1 pW to ~4 μW). The SiPM has a manageable bias voltage (~25 V to ~30 V DC) for systems integration, and with optimised sensitivity it will enhance bio-photonic research in the area of AF to detect intraoperatively, for example, brain tumour margins.

## 1. Introduction

Bio-photonics uses light–tissue interactions such as reflectance, scattering, absorption, time-of-flight (ToF) and fluorescence to evaluate the properties and composition of tissues [1]. These diagnostic techniques can be performed on in vivo and ex vivo tissue through minimally invasive procedures, offering attractive alternatives to traditionally used medical diagnostic tools and also enabling research for the next generation of medical devices [2,3]. These devices are traditionally optical fibre based and integrated into a catheter or guidewire for intra operative use for surgical guidance and diagnostics [3]. Fibre-coupled devices in particular can image deep inside the body and may have large source detector distances. They require ideal sensors able to detect light powers in the pico-watts or sub-pico-watts range and for ToF applications to have a time resolution in the order of 100 pico-seconds. Designing a high-sensitivity, wide dynamic range and low-noise application-specific integrated circuit (ASIC) for silicon photomultipliers (SiPMs) with the potential for device heterogeneous integration, miniaturisation and surgical tool integration will enhance clinical utility. This integrated ASIC and SiPM will have particular application for brain tumour margin resection using auto-fluorescence (AF) [3,4,5]. Additionally, use of multispectral and hyperspectral optical source techniques for deep tissue measurements and AF measurements for biomarker detection further challenges the sensitivity and dynamic range requirements of detectors and their supporting electronics [2,4,6].

The majority of the recent bio-photonics research omits detailed description of their detection systems [7]. However, some published works contain detailed discussion of the photodetectors employed. Some of the research listed here used photodiodes (PDs) [8,9], avalanche photodiodes (APDs) [2,4,10,11,12,13,14], single photon avalanche photodiodes (SPADs) [15], and silicon photomultipliers [16,17,18,19,20]. These works primarily focus on the specific bio-photonics applications rather than discussing the specifications of the photo detection devices.

This work seeks to build on this by exploring current state-of-the-art SiPMs. Most bio-photonics measurement set-ups are bulky, complex and expensive. SiPMs are highly suitable for deployment outside the lab due to their low cost, small surface area, high sensitivity and integration capabilities [21,22,23]. This paper is organised as follows: Section 2 presents a short review of the different light measurements used in bio-photonics, and Section 3 describes the SiPM. Section 4 and Section 5 focus on a comparison between the SiPM and other photodetectors and their impact on the readout electronics. Section 6 presents some measured results [2]. Section 7 summarises the work and presents conclusions.

## 2. Bio-Photonics

In this section, three main bio-photonics signal measurements are presented: diffuse reflectance spectroscopy (DRS), auto-fluorescence (AF) spectroscopy and Raman spectroscopy (RS).

DRS is a technique that measures a diffuse reflectance spectrum of tissue. The shape of the obtained spectrum is primarily determined by the absorption and scattering properties of a sample. The technique is used to obtain the optical properties of a sample at different wavelengths. Reference spectra can be used to compare deviations in the response, such as discriminating healthy tissue from cancerous tissue. DRS has shown outstanding results, achieving up to 94% specificity and sensitivity in ex vivo liver biopsy clinical trials, discriminating cancerous and healthy liver tissue [24]. Tanis et al. performed DRS measurements in an in vivo setting by means of an optical needle, accurately discriminating cancerous colorectal tissue with a sensitivity of 95% and specificity of 92% [25]. Other studies involving DRS include colorectal cancer [6,26,27], breast cancer [28] and oral cavity cancer [29].

In in vivo optical fluorescence imaging, molecular dyes, such as photosensitising agents, are commonly used to identify the different biological processes and mechanisms in the body [30]. One agent frequently used is 5-amino levulinic acid (5-ALA). When administered to the body it will promote the expression of the fluorescent substance protoporphyrin IX (PpIX), particularly in cancer cells, allowing for the detection of tumours. However, such agents are expensive and are not certain to fluoresce in all patients upon light excitation [31,32]. Furthermore, if a fluorophore is excited for too long, photo-bleaching may occur, in which case the molecule is altered, and the experiment cannot be carried out [33].

AF, also known as “native fluorescence”, eliminates such disadvantages, as it is a naturally occurring phenomenon. AF spectroscopy measures the emissions of endogenous fluorophores—fluorophores originating from the body [34,35]. In the case of AF, the tissue is excited with light at specific wavelengths and re-emits light at longer wavelengths. For this reason, one can look for the presence of specific AF for biomarkers—molecules formed in the body as a response to a disease. These biomarkers can be differentiated from one another due to their different Stokes-shifted emission at different wavelengths. The 5-ALA PpIX system produces strong emissions, whereas most AF emissions require very sensitive detectors depending on the biomarker.

DRS can be described as a broadband technique, while AF is typically assessed wavelength-specific in that testing and analysis are made on the basis of a narrowband wavelength of incident light. This makes these techniques complementary optical measurements. For this reason, many studies have combined both techniques for tissue diagnostics, using the cross-correlated signals to obtain both specificity and sensitivity [36,37,38]. Lu et al. investigated the use of DRS and AF to differentiate between healthy grey matter and brain tumours. Findings showed a sensitivity of 89.3% and a specificity of 92.5% [2].

RS is the bio-photonics technique that offers the highest molecular specificity using inelastic scattering of photonics to determine the molecular vibrational modes. RS is non-destructive and label free but typically requires a narrow bandwidth laser and a highly sensitive spectrometer [39,40,41]. The structural and chemical information of molecules can be determined by observing the peak position and relative peak intensity of the spectrum. Majumder et al. presented a study comparing the accuracy of DRS, AF and RS in classifying healthy breast tissues and three different types of tumours [42]. It was found that RS provided the greatest accuracy, close to 99%. A review of the principles of RS and recent advances is presented by Krafft et al. [43]. Aubertin et al. used RS to distinguish the prostate and the surrounding tumorous tissue with a sensitivity of 82% and a specificity of 83%. Moreover, benign tissue was differentiated from malignant tissue with a sensitivity of 87% and specificity of 86% [44]. In the work presented, no RS measurements were performed; however, the high sensitivity of the SiPMs at low light levels suggest that they are a great candidate for RS.

In bio-photonics, the light intensity measurements vary over greater than six to seven orders of magnitude, but the exact detectable light power is difficult to measure or quantify for each optical phenomenon in general. Table 1 below shows a summary of the different optical techniques and their typical optical power ranges.

## 3. The Silicon Photomultiplier

Modern photodetectors operate on the principle of the photoelectric effect—emission of electrons in response to a stimulus—to convert optical signals into electrical signals, which are proportional to the incident optical power at a given wavelength. The bandgap energy of a semiconductor represents the amount of energy required to promote a valence electron—an electron that is bound to an atom—to become a conduction electron, a charge carrier that is free to move within the material and, hence, conduct current. As light interacts with the photodetector, photons of energy greater than that of the bandgap energy of the semiconductor excite the electrons in the structure, causing them to transition from the valence band to the conduction band. This process, called impact ionisation, causes the carriers to kick some of the electrons in the materials out of the valence band into the conduction band, forming additional electron–hole pairs. This leads to a sustainable flow of current through the semiconductor when an electric field is applied across it.

The SPAD is a photodetector that is operated beyond its reverse bias breakdown voltage, V_BD_, in a metastable state known as the Geiger mode. When a single photon is detected, a large exponential current flows through the SPAD, providing a short trigger pulse output. SPADs can also be used to obtain the time of arrival of each photon due to their fast avalanche current generation.

Figure 1 illustrates the operation of the SPAD.

Initially, the reverse voltage of the device exceeds its breakdown voltage by an amount V_OV_, the overvoltage or excess voltage; see Figure 1a. Typically, V_OV_ is 10 to 25% higher than the breakdown voltage. While no photon is detected, the device is in a metastable state, and there is no current flow. When a photon is detected, as seen in Figure 1b, a large current flows through the device. However, in order to prevent the device from getting damaged and to allow for the detector to detect a subsequent photon, the avalanche needs to be stopped. This process is known as quenching, as shown in Figure 1c, and it is achieved by bringing the bias voltage back down to the breakdown voltage by the use of a resistor or transistor. This quenching device can be followed by an inverter to create a single digital pulse for each single photon detected. The time taken for the SPAD to recharge is known as its dead time. During this recharge phase, no photons can be detected.

Bruschini et al. provided a full review of SPAD imagers for bio-photonics applications [45]. Our review does not include ToF applications; however, SPAD ToF applications can be found in [46].

While the output of the SPAD is binary, either providing an ‘on’ or an ‘off’ state, an alternative analogue device based on SPAD cells called the silicon photomultiplier (SiPM) can provide information on the magnitude of the instantaneous photon flux. The SiPM integrates an array of independent microcells, each comprised of a SPAD sensor and its own quenching resistor, as shown in Figure 2.

This structure allows for any cell to detect an incoming photon and recharge while the other cells remain charged and ready to detect another photon [47]. The output values of all microcells are summed together, providing a quasi-analogue output, which is proportional to the instantaneous photon flux. Piemonte et Gola presented a full review of SiPM and their advances [48], and Klanner provided methods to characterise SiPMs [49].

**Figure 2 biosensors-12-00793-f002:**
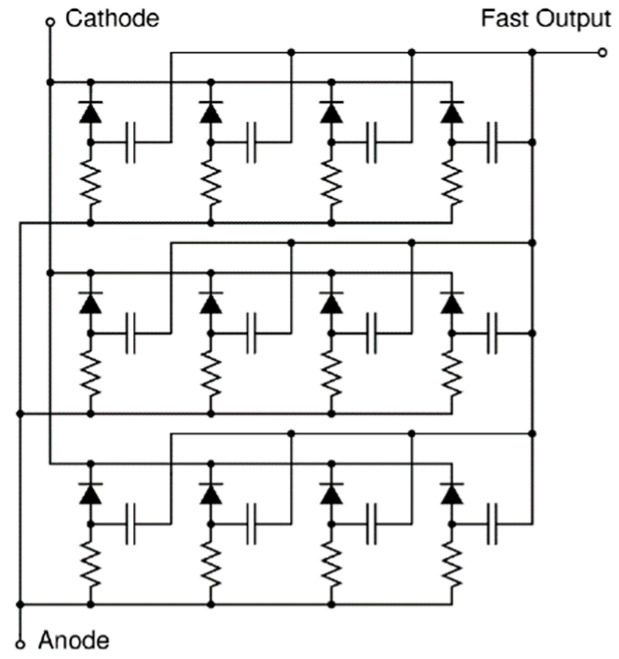
Simplified schematic of the SiPM from the supplier OnSemi, consisting of an array of SPADs and quenching resistors. Reprinted with permission from ref. [50]. Semiconductor Components Industries, LLC. (Phoenix, AZ, USA), dba onsemi 2022.

The gain of a SiPM is presented as the amount of charge generated per detected photon, and it is a function of both the microcell size and the overvoltage V_OV_. Since each cell generates a quantised amount of charge for each event, the gain of each independent microcell is defined in (1).
(1)$$G=\frac{C_{d}V_{\mathrm{OV}}}{q}\;$$
where $C_{d}\;$ is the microcell capacitance, and q  is the electron charge. Overvoltage limits are stated in the SiPM data sheets. The relationship between the overvoltage and the resulting gain of the device is illustrated in Figure 3 for three different OnSemi SiPMs; each device has the same area (1 mm × 1 mm) but different microcell sizes: 10 μm, 20 μm and 35 μm. As expected, the gain is proportional to the microcell size due to the increasing microcell capacitance, $C_{d}$.

The photon detection efficiency (PDE) of a SiPM is defined as the probability that a single photon of wavelength λ is absorbed and creates a flow of electron–hole pairs (i.e., photocurrent) that results in an output pulse. The PDE is defined as follows:(2)$$PDE\left(\lambda ,\;V_{OV}\right)=\eta \left(\lambda \right)\times \varepsilon \left(V_{OV}\right)\times FF$$
where η(λ) is the quantum efficiency (QE) of silicon—the ratio of photo-carriers produced to the number of incident photons; it is a function of wavelength. $\varepsilon \left(V_{OV}\right)$ is the avalanche initiation probability, which itself is dependent on the overvoltage, and FF is the fill factor of the SiPM—the ratio of the sensitive (or active) area to the overall area of the device [51]. Figure 4 shows the PDE of the SensL (OnSemi) SiPM device, MicroFC−30035−SMT. The size of the active area is (3 mm × 3 mm), and the microcell size is 35 μm. The PDE is presented for two different overvoltages, showing that similarly to the gain of the device, the PDE is highly dependent on the overvoltage as well as the wavelength.

While the SiPMs discussed in this review only presented OnSemi devices, Table 2 highlights other manufacturers. The authors only included low area devices.

## 4. Detector Performance Comparison

Table 3 highlights the differences between three types of detectors used in bio-photonics applications: a PIN diode, an APD and a SiPM. The SiPM has several advantages compared to the other two detectors. The SiPM presents the lowest area of the three detectors while keeping a high active area to a total area ratio, fill factor (FF), of 42%. Moreover, the SiPM has the highest responsivity: the number of carriers generated due to the absorption of incoming photons. This, in turn, directly produces a higher photocurrent, as the following relationship applies:(3)$$i_{in}\left(t\right)=RP_{in}\left(t\right)$$
where *i_in_* is the photocurrent, *P_in_* is the incident optical power, and *R* is the responsivity of the detector, described as follows:(4)$$R=\frac{\eta q}{h\nu }$$
where *η* is the quantum efficiency, and *hν* is the photon energy. Hence, the SiPM can produce a higher photocurrent than other types of detectors, leading to higher resolution and SNR. While the performance of PIN diodes and APD can easily be described in terms of quantum efficiency, (2) shows that this metric differs in the case of SiPMs. Due to their different structure, i.e., the presence of multiple SPAD arrays, the PDE is highly dependent on the microcell size, which in turns affects the FF of the device. When the size of the device remains unchanged and the microcell size is increased, the FF is increased and the PDE rises, as seen from (2).

When measuring low optical power, the photocurrent produced is in the order of nano-amperes. For this reason, the noise introduced by the photodetector needs to be kept low, while the signal-to-noise ratio (SNR) needs to be high. Two main sources of noise in PIN diodes are shot noise and the dark current noise. Shot noise arises due to the random characteristics of photon detection and is seen as fluctuations in the electric current produced by the detector itself. In photodiodes, the dark current is a leakage current; it flows regardless of whether illumination is present or not. The fluctuation in this dark current is a random noise created by random thermally generated electron–hole pairs and contributes to the overall photocurrent. The dark current can be reduced by lowering the bias voltage of the photodiode and controlling the temperature of the device.

Shot noise and dark current noise are also present in APDs. However, an additional multiplication factor called the excess noise factor is added to the dark current, an effect which arises from the avalanche generation process. In the process of avalanche multiplication, ionised charge carriers travel through the multiplication region of the detector at very high velocity and collide with other atoms in the lattice, generating electron–hole pairs. The process of generating new electron–hole pairs is statistical and highly depends on the detector construction, the electric field of the device and the ionisation rate of the electrons and the holes.

The noise sources described above also apply to the SiPM; however, additional noise sources are found in SiPMs. The largest source of noise in SiPMs is the dark count rate (DCR), which differs from the dark current. This signal is uncorrelated to the arrival of a photon; it is created by thermally excited electrons that can initiate an avalanche event and cannot be distinguished from a real photon absorption event. Other sources of noise include correlated noise, namely, optical crosstalk and after-pulsing [57]. Optical crosstalk between microcells occurs during the avalanche multiplication, where charge carriers can emit a photon in the near infrared (NIR) region, which can travel through the silicon and reach other microcells, causing a subsequent avalanche. In SiPMs, crosstalk can be detected at the output, as the magnitude and shape of the SiPM varies depending on the number of coincident pulses. After-pulsing is a phenomenon that arises due to the spontaneous release of trapped charges in the silicon after photocurrent generation. Due to the statistical nature of the photocurrent generation, the excess noise factor is used to summarise these two noise contributions, crosstalk being the prevalent source [58]. The noise factor of the SiPM is much less than that of photomultiplier tubes (PMTs) and APDs [59].

As presented in Table 3, the noise floor of the detectors were used to determine the minimum optical power that can be detected by each respective device. Similarly, the maximum detectable optical power was set to that of the saturation limit of the device. It is worth noting that for the SiPM, the maximum optical power used for calculations was 1 μW, as opposed to the data sheet maximum value of 4 μW, to ensure linearity in the response.

Other sources of noise can be found in photo-detection systems, such as photon noise, ambient light interference and electronics circuit noise [60]. Photon noise arises from the random fluctuations of light intensity in the laser light source as well as the internal reflection in the optical system. In the detection, there is a number of processes generating dark signals. These signals can generally be subtracted, while the uncertainty is noise. This includes the statistical uncertainty in the number of photons detected in a certain time interval—known as photon or Poisson noise. Ambient light interference occurs when the detector is poorly isolated from ambient light. To cancel ambient noise, a background measurement is normally collected prior to other measurements, and subtraction is performed post-processing to remove its effect from the measured data.

Amongst the available photodetectors, the SiPM offers the best timing resolution, an ideal feature for fast measurements such as AF. Referring to Table 2, we note that the peak detection wavelength of SiPMs varies from one device to another, allowing for bespoke device requirements. The detection of nicotinamide adenine dinucleotide (NADH) is widely used in bio-photonics applications—and in our current research; its photoluminescence spectrum is shown in Figure 5. Noting a peak fluorescence at 476 nm, the SiPM chosen in Table 3 fits our requirements. In the case of brain tumours, Saraswathy et al. have shown that a wavelength of 470 nm was optimal for AF measurements [61].

## 5. Readout Circuit Comparison

The photocurrent produced by the photodetectors is converted to a voltage using a trans-impedance amplifier (TIA), as shown in Figure 6. A TIA is a current to voltage converter, where the output voltage is given by
(5)$$V_{out}=i_{in}R_{TIA}$$
where *i_in_* is the photocurrent flowing from the detector upon light detection, and *R_TIA_* is the resistive gain of the TIA. The voltage noise produced by the resistance *R_TIA_* can be seen as a voltage source *V_noise_* placed at the output of the TIA, hence adding an error voltage to the conversion, as seen in Figure 6. As the noise associated with this error voltage cannot be removed at the post processing stage, it is important that minimising its effect be considered at the design stage.

In order to further compare the performance of the detectors presented in Section 3, we computed which gain *R_TIA_* was required to scale the maximum photocurrent of each detector to a fixed voltage output of 4 V. For this reason, the noise introduced by the feedback resistance was represented as a voltage source as opposed to a current source as the voltage swing was maximised without saturating the detector. Due to the inherently large gain of the SiPM, a smaller resistance gain *R_TIA_* is required, which, in turn, reduces the amount of thermal noise added to the sensed signal by the current to voltage conversion. The thermal voltage noise density of a resistor is
(6)$$\bar{V_{noise}^{2}}=4kTR_{TIA}$$
where *k* is Boltzmann’s constant, and *T* is the temperature. Table 4 summarises the different gains required for the current to voltage conversion stage for each detector, as well as its resulting additional noise.

## 6. Use of Care: Tissue Recognition Project

In this section, a bio-photonics tissue-recognition biopsy device is discussed. A previous version of the work by Lu et al. [4] is presented, and its adaptation into a board with SiPMs is demonstrated.

Aiming to develop a bio-photonics tissue-recognition device for surgical guidance in vivo, Lu et al. investigated the introduction of multiple illumination and detection bands [4]. Specifically, eight illumination sources and eight APDs were used to simultaneously collect DRS and AF spectra to distinguish between healthy and cancerous tissue. The instrumentation used and tested by Lu et al. is presented in Figure 7 [2]. The system was comprised of a light source with eight fibre-coupled LEDs, with wavelengths ranging from 300 nm to 700 nm, and a fibre optic bundle. The light collected from the tissue was split into spectral bands and sent to an optical assembly through the use of dichroic mirrors and band-pass filters. The intensities of the bands were detected by the APDs associated with each wavelength. The voltage output of the APDs was integrated using a low-pass filter (LPF) and amplified using an eight channel dynamic amplifier. Because of the difference in amplitude of DRS and AF signals (see Table 1), a digital output module was used to set the gain in each of the channels to 1 for DRS and 100 for AF measurements. Signals were sampled at 30 kHz and digitised using a NI-9205 data acquisition card.

To reduce the loss of light intensity through the system, each detector has its own circuit, comprising a TIA and an LPF. This implementation defines a system that has eight different detection channels, eight identical sub-circuits and eight band-pass filters.

In this work, we developed a single-board system with similar functionality based on SiPMs. A printed circuit board (PCB) was produced containing these different measurement channels, each comprised of a SiPM from ON Semiconductor (MicroFC-10010-SMT), a TIA (AD8627) and a LPF (THS4551). The detectors were placed at the centre of the board, allowing for the fitting of a casing containing the optical components, as seen in Figure 8a. The circuit for each channel is shown in Figure 8b. The low-pass filter has a gain of 1.5 and a cut-off frequency of 100 kHz.

While the dynamic range of the OnSemi C-10010 SiPM is rated from ~1 pW to ~4 μW of input optical power, the relationship between the light and the observed photocurrent is non-linear, as shown in Figure 9. To explain this non-linearity, (7) describes the relationship between the number of incident photons, *N_photon_*, and the number of microcells fired as a response to incoming light, *M_fired_*. This shows that *M_fired_* is highly dependent on the PDE, which itself varies with wavelength and overvoltage and the total number of microcells in the device, *M_total_*.
(7)$$M_{\mathrm{fired}}=M_{\mathrm{total}}\left(1-e^{\frac{-PDE\times N_{\mathrm{photon}}}{M_{\mathrm{total}}}}\right)$$

For a SiPM of the same area, we note that increasing the microcell size drastically reduces the dynamic range of the device. For example, in a 1 mm × 1 mm device, if the size of the microcells is 10 μm × 10 μm (amounting to 2880 microcells), the maximum detectable optical power is ~4 μW. On the other hand, if the microcell size is increased to 35 μm × 35 μm, the total number of cells is reduced to 504, lowering the maximum detectable power to ~20 nW.

When designing the readout electronics, the gain, *R_TIA_,* must be chosen so as to remain in the linear region of the device, by-passing the exponential response of the SiPMs, as stated in [59]. While the gain of the TIA affects the dynamic range, the overvoltage plays a more important role in defining the complete photon to voltage conversion. Figure 10a shows the output response of the circuit for three different overvoltages, keeping the gain factor *R_TIA_* constant. Increasing the overvoltage increases the gain, as well as the PDE of the SiPM, making the device saturate at lower optical powers. However, to prevent this, Figure 10b shows that for the same overvoltage, reducing the feedback resistance *R_TIA_* increases the dynamic range of the system. On the other hand, lower values of *R_TIA_* increase the sensitivity of the system at low incident light power, allowing for a better resolution, requiring less sensitive analogue to digital converters. Moreover, the noise introduced in the circuit is lowered, as seen from (6).

In order to obtain the optical dynamic range of the board presented in Figure 8a, measurements were performed using an optical fibre pointed at the SiPMs under test and neutral density (ND) filters of different optical densities (OD) to attenuate the intensity of the incident optical power. The incident light was generated using a 470 nm Thorlabs M470F3 fibre-coupled LED to a 200 nm fibre with numerical aperture of 0.22. The optical power coming out of the fibre was recorded prior to measurements, and its intensity was reduced by placing combinations of ND filters. The attenuated optical power was calculated from the combination of filters verified against the theoretical response of the board. Figure 11 shows measured versus simulated results. The objective of this experiment was to demonstrate the linear response of the detector at low-light powers. No post-processing was applied to these data. At ambient light, the SiPMs were saturated, and the power drawn by the detectors was only 0.435 mW, while the power drawn by the rest of the electronics on the board was 0.498 mW.

In this work, we have successfully adapted an existing light detector system, shown in Figure 7, reducing its benchtop size to a single PCB (30 mm × 70 mm). Using Equation (7), the photocurrent produced in response to a given optical power can be computed. From that, the feedback resistance can be scaled to match a given range of input light power. At a wavelength of 470 nm, a feedback resistance, *R_TIA_* = 7 kΩ, offers a light swing ranging from ~1 pW to 1 nW, while a resistance of 330 Ω offers a light swing ranging from ~1 pW to 250 nW. By adapting the gain of the TIA, the sensitivity of the system at low optical power can be improved. It is also noted that the performance of this system will be further improved with the use of repeated measurements and algorithms as well as calibration.

## 7. Conclusions

Careful selection of the nominal overvoltage, TIA gain resistance and circuit layout can allow PCB mounted SiPM detectors to achieve a considerable dynamic range at very low optical power, superior to PIN diode- and APD-based systems. Moreover, we have shown the SiPM to be suitable in area- and power-constrained applications while allowing a manageable bias voltage in the range of 30 V and minimal dark current. The spectral response of the SiPM is also ideal for AF measurements in the visible part of the spectrum. Despite single photon counting being a better measurement technique for RS, this work only discusses photocurrent integration methods, as single photon counting has a limited dynamic range and hence is not suitable for DRS and AF measurements. For this reason, current integration mode is a better readout method to enable measurements of all three signals.

Future work will further investigate the limitations of the design presented with respect to maximizing the sensitivity to 1 pW level of incident optical power, improving further and adjusting the dynamic range of the system while maintaining a linear response. The presented work will be integrated as an ASIC to further reduce the system’s size and noise, improving its performance. This work demonstrates that the SiPM is a suitable detector for bio-photonics measurements, such as enabling AF for brain tumour margin resection.

## Figures and Tables

**Figure 1 biosensors-12-00793-f001:**
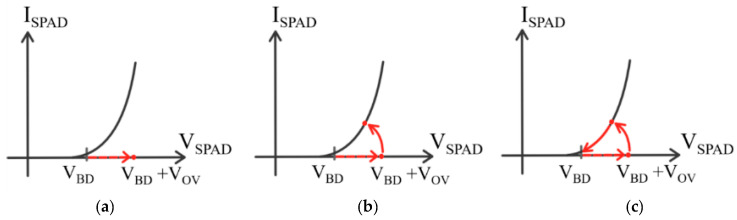
Operation of the SPAD, variation in overvoltage demonstrated in red. (**a**) Device in metastable state, waiting for a photon to hit the surface. (**b**) A photon was detected and a large current flows through the device. (**c**) Quenching is realised, and the bias voltage is brought back to the breakdown voltage. Once the SPAD is recharged, the bias is increased to V_BD_ + V_OV_ and another photon can be detected.

**Figure 3 biosensors-12-00793-f003:**
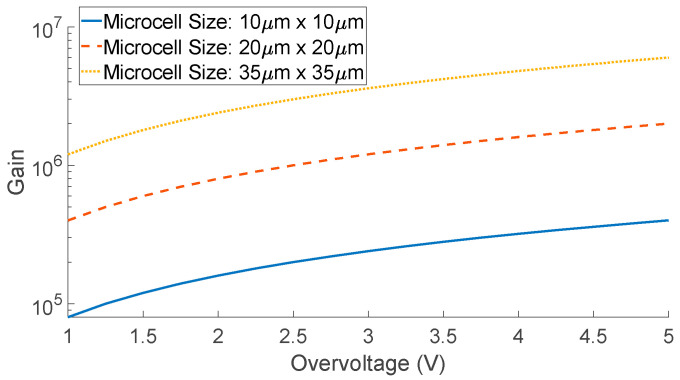
Gain vs. overvoltage of three SiPMs of the same area (1 mm × 1 mm) but different microcell sizes: 10 μm, 20 μm, and 35 μm.

**Figure 4 biosensors-12-00793-f004:**
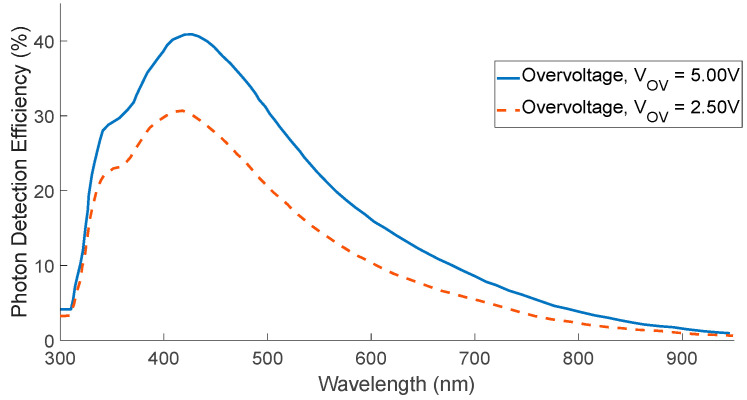
PDE versus wavelength for the OnSemi C-Series, 3 mm × 3 mm, 35 μm microcell size, for two different overvoltages. Reprinted with permission from ref. [50]. Semiconductor Components Industries, LLC., dba onsemi 2022.

**Figure 5 biosensors-12-00793-f005:**
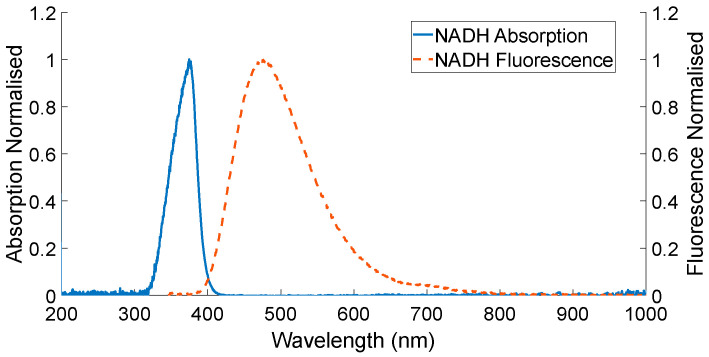
Photoluminescence spectrum of NADH.

**Figure 6 biosensors-12-00793-f006:**
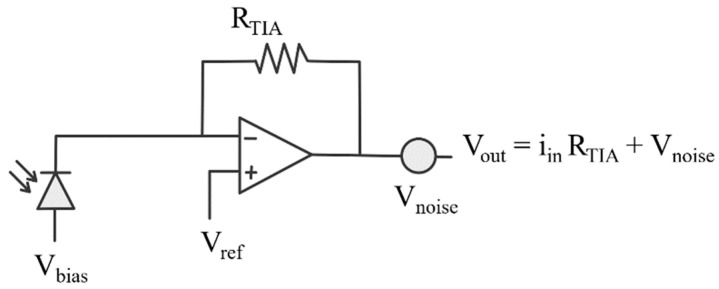
Photocurrent to voltage conversion stage including noise due to feedback resistance.

**Figure 7 biosensors-12-00793-f007:**
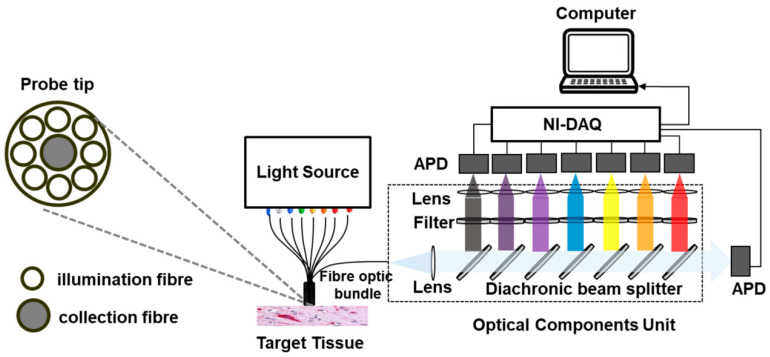
Instrumentation used in the Tissue Recognition project. Reprinted with permission from [2]. © The Optical Society 2022.

**Figure 8 biosensors-12-00793-f008:**
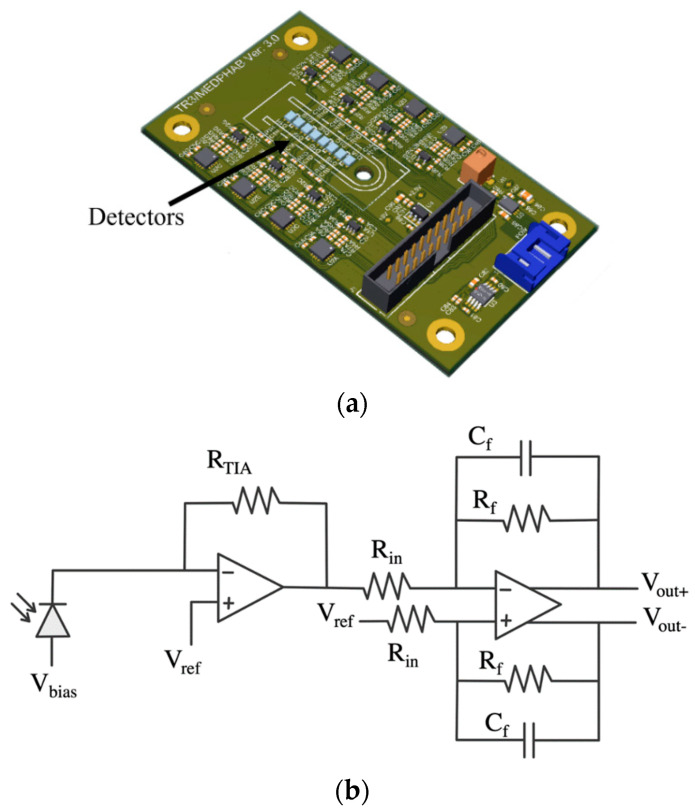
(**a**) PCB of the detection system. Each SiPM has its own TIA and LPF. Detectors are placed in the centre of the board, allowing for the fitting of opto-mechanical components: micro-filters and casing. (**b**) Schematic of one of the channels on the PCB. Input is the photocurrent flowing from the SiPM, turned into a voltage with the TIA, and the filtered stage is differential, removing common mode noise.

**Figure 9 biosensors-12-00793-f009:**
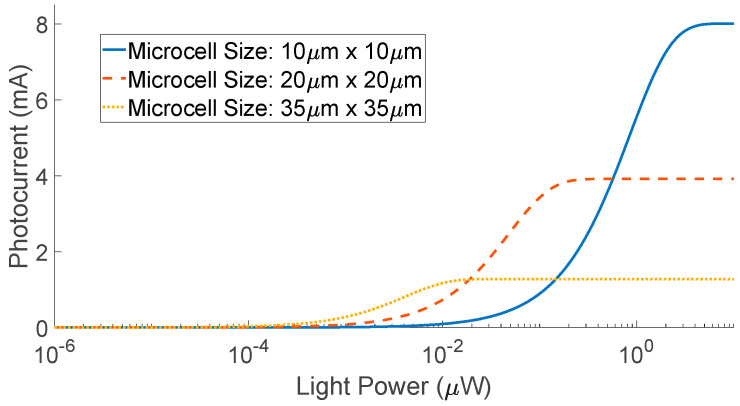
Input optical power versus corresponding photocurrent generated by the OnSemi C−Series (1 mm × 1 mm) with different microcell sizes, 10 μm, 20 μm and 35 μm, respectively. The total number of cells in each device is 2880, 1296 and 504, respectively.

**Figure 10 biosensors-12-00793-f010:**
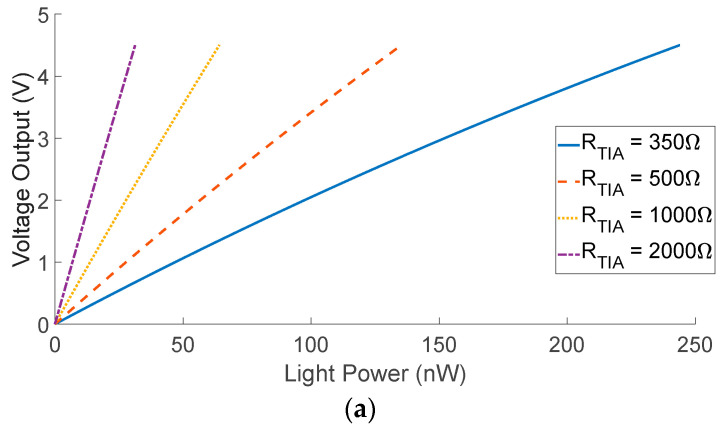
(**a**) Voltage output response of the system. The overvoltage V_OV_ is varied for three different values and *R_TIA_* = 1.5 kΩ. (**b**) Voltage output response of the system. The feedback resistance, *R_TIA_* is varied for four different values. The overvoltage is kept constant, V_OV_ = 5 V.

**Figure 11 biosensors-12-00793-f011:**
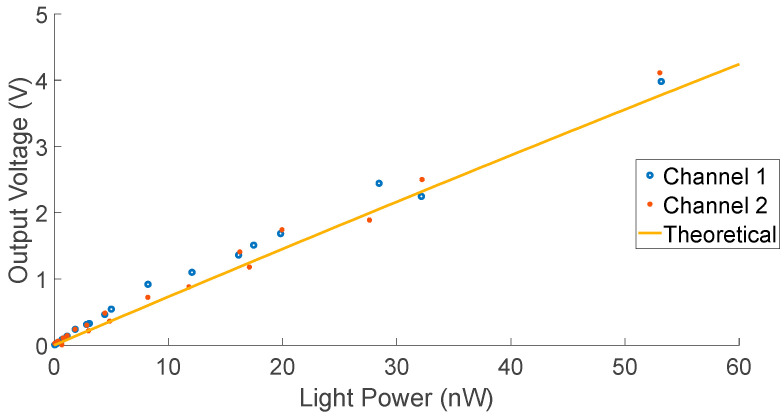
Measured results. Channel 1 and channel 2 are two different SiPMs on the same board. *R_TIA_* = 1.5 kΩ and *V_OV_* = 3.6 V.

**Table 1 biosensors-12-00793-t001:** Optical phenomena and optical powers.

	Diffuse Reflectance	Auto-Fluorescence	Raman Spectroscopy
Optical Power (W)	1–10 μ	10–100 n	1–10 p

**Table 2 biosensors-12-00793-t002:** Comparison of different SiPM devices.

	OnSemi [50] (MicroFC-10010)	Hamamatsu [52] (S14160-1310PS)	Hamamatsu [52] (S14160-1315PS)	AdvanSid [53] (ASD-RGB1S-P)	AdvanSid [54] (ASD-NUV1S-P)
Gain	2 × 10^5^	1.8 × 10^5^	3.6 × 10^5^	2.7 × 10^6^	3.6 × 10^6^
Operational Bias (V)	24.2–24.7	35–41	35–41	27–29	26–28
Overvoltage (V)	1–5	5	4	2–4	2–6
Spectral Range (nm)	300 to 950	290 to 900	290 to 900	350 to 900	350 to 900
Peak Sensitivity (nm)	420	460	460	550	420
PDE (%)	14	18	32	32.5	43
Cell Capacitance (fF)	12.8	5.8	14.4	90	90
Area (mm × mm)	2.394	5.523	5.523	5.034	5.034
Active Area (mm × mm)	1 × 1	1.3 × 1.3	1.3 × 1.3	1 × 1	1 × 1
Microcell Size (μm)	10 × 10	10 × 10	15 × 15	40 × 40	40 × 40

**Table 3 biosensors-12-00793-t003:** Performance comparison of the PIN diode, APD and SiPM.

	PIN [55] (SFH2704)	APD [56] (S12053-05)	SiPM [50] (C10010)
Gain	1	1–50	2 × 10^5^
Output Type	Analogue	Analogue	Analogue or Digital
Operational Bias (V)	6	150–200	24.2–24.7
Overvoltage (V)	–	–	1–5
Spectral Range (nm)	400 to 1100	200 to 1000	300 to 950
Peak Sensitivity (nm)	900	620	420 *
PDE/QE (%)	–	80	18 **
Capacitance (pF)	13.4	5	50
Max Photocurrent (μA)	1.22	84	16 × 10^3^
Dark Current (nA)	0.1–25	0.2–5	1–10
Area (mm^2^)	3.6	21.24	2.4
Active Area (mm^2^)	1.51	7.07	1
Responsivity (A/W)	0.34	21	4 × 103
Rise Time (ns)	47	0.875	0.3
Min Detection Optical Power (pW)	73.53 × 10^3^	399	2.5
Max Detection Optical Power (μW)	4.41	16	1
Optical Dynamic Range (dB)	17.78	46.03	56
Single Photon Detection	No	No	Yes
Photon Time Stamping	No	No	Yes

* Peak wavelength is particular to the C-Series model. ** PDE is higher in larger devices and/or higher *V_EX_*.

**Table 4 biosensors-12-00793-t004:** Current to voltage conversion noise contribution for each photodetector.

	PIN [55] (SFH2704)	APD [56] (S12053-05)	SiPM [50] (C10010)
Max Photocurrent (μA)	1.22	126	8 × 10^3^
TIA Resistance (Ω)	1.23 × 10^6^	11.91 × 10^3^	187.5
TIA Output Noise Density (nV/$\sqrt{Hz}$)	142.69	14.04	1.76

## Data Availability

Not applicable.

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
