# Peer review of "Silicon Photomultiplier—A High Dynamic Range, High Sensitivity Sensor for Bio-Photonics Applications"

_biosensors, 2022, doi:10.3390/bios12100793_

Round 1

Reviewer 1 Report

1. Title and abstract of the paper supposed to be reframed to narrate the high sensitivity of the proposed sensor.   

2. Figure 3 is not described in the text (see section 3). 

3. Reframe the heading titles of section 4, 5 and 6.  

4. Conclusions are supposed to reflect the potential outcomes of the intended work as well as inline with the abstract of the article. Therefore, it is advised to revise this section. 

5. In Eq. (1) VEX is termed as overvoltage while earlier VOV has introduced as overvoltage. Justify the choice of notation and their intended function. In Eq. (2), PDE is narrated as a function of wavelength and voltage while in Eq. (7) it is described as a function of wavelength and overvoltage (in lines 367-369). Rectify the discrepancy. 

6. Define PMT as introduced in the line 240.

7. Cite a suitable reference in line 315. Correct the spelling mistake at various places (for example, check the lines 78 and 297). 

8. Define SiPM and AF once in line 36 and 38, respectively, where they have appeared at first.

9. TIA Output Noise Density is the key factor to control the overall performance of SiPM. It is advised to describe it in more details in section 5 (if possible) for the sake of readers.  

10. Define "On Semi SiPM" as mentioned in figure caption of Fig. 2, where it is introduced at first.     

11. It is suggested to use lines of different styles in Figs. 3, 5, 9, and 10 in order to distinguish them in black and white printout. 

12. Microcell size of 50 micron is reported (in the figure caption of fig. 9) while in figure legend maximum size of 35 micron is depicted. Rectify the discrepancy.

13. Authors has reported the maximum detectable light power of the device from ~4μW to ~20nW respectively (in lines 374) while in lines 420-425 they have reported powers ranging from a few pW to 240nW. Correlate these values; moreover, comment on the choice of feedback resistance of ~400Ω.   

14. It is advised to reframe the text related to Figure 11 (regarding the measured and simulated results) in order to understand the aim of the anticipated work.

15. Use uppercase for author's name in Ref. [61]. It is advised to be particular in using the words such as while and whilst.

Author Response

Dear Reviewer,

Thank you for taking the time to review this submission, your comments and suggestions are much appreciated.

In response to your comments and suggestions, please find the following:

  1. Title and abstract of the paper supposed to be reframed to narrate the high sensitivity of the proposed sensor.   

Changes were applied to the manuscript.

  1. Figure 3 is not described in the text (see section 3). 

Changes were applied to the manuscript.

  1. Reframe the heading titles of section 4, 5 and 6.  

Changes were applied to the manuscript.

  1. Conclusions are supposed to reflect the potential outcomes of the intended work as well as inline with the abstract of the article. Therefore, it is advised to revise this section. 

Changes were applied to the manuscript.

  1. In Eq. (1) VEXis termed as overvoltage while earlier VOVhas introduced as overvoltage. Justify the choice of notation and their intended function. In Eq. (2), PDE is narrated as a function of wavelength and voltage while in Eq. (7) it is described as a function of wavelength and overvoltage (in lines 367-369). Rectify the discrepancy. 

Changes were applied to the manuscript. The authors chose to use the term VOV was chosen.

  1. Define PMT as introduced in the line 240.

Changes were applied to the manuscript.

  1. Cite a suitable reference in line 315. Correct the spelling mistake at various places (for example, check the lines 78 and 297). 

Changes were applied to the manuscript.

  1. Define SiPM and AF once in line 36 and 38, respectively, where they have appeared at first.

Changes were applied to the manuscript.

  1. TIA Output Noise Density is the key factor to control the overall performance of SiPM. It is advised to describe it in more details in section 5 (if possible) for the sake of readers.  

Changes were applied to the manuscript.

  1. Define "On Semi SiPM" as mentioned in figure caption of Fig. 2, where it is introduced at first.     

Changes were applied to the manuscript.

  1. It is suggested to use lines of different styles in Figs. 3, 5, 9, and 10 in order to distinguish them in black and white printout. 

Changes were applied to the manuscript.

  1. Microcell size of 50 micron is reported (in the figure caption of fig. 9) while in figure legend maximum size of 35 micron is depicted. Rectify the discrepancy.

Changes were applied to the manuscript.

  1. Authors has reported the maximum detectable light power of the device from ~4μW to ~20nW respectively (in lines 374) while in lines 420-425 they have reported powers ranging from a few pW to 240nW. Correlate these values; moreover, comment on the choice of feedback resistance of ~400Ω.   

The reported maximum light powers reported in line 384 were directly related to the change in microcell size for a detector of the same area. This has been made clearer in the reviewed manuscript.

In the discussion from line 448 onwards, the newly reported light power range depends on the sizing of the feedback resistance. Using equation (7), the photocurrent produced in response to a given light power can be computed. From that, the required TIA resistance can be calculated. For example, if we want to allow a light power swing ranging from ~1pW to 250nW at a wavelength of 470nm, a feedback resistance of 330 should be used. For light powers from ~1pW to 500nW, a resistance of 190 should be used. As seen in Figure 10 a) this allows us to use the detector in its linear region, as well as optimise the dynamic range of the system.

  1. It is advised to reframe the text related to Figure 11 (regarding the measured and simulated results) in order to understand the aim of the anticipated work.

Changes were applied to the manuscript.

  1. Use uppercase for author's name in Ref. [61]. It is advised to be particular in using the words such as while and whilst.

Changes were applied to the manuscript.

Kind regards,

Rachel Georgel

Reviewer 2 Report

 This work is an overview of silicon photomultipliers with a view to defining their importance for bio-photonic and clinical applications. The results show that careful selection of gain-determining components and integration of the device with micro-optics can allow the detector to achieve significant sensitivity for automatic detection at low light powers.

The subject itself is appropriate for this journal, and the results are appropriate. However, there are several issues as will be explained in below:

1. It is better to revise the text once again and correct typographical errors. For example:

“utilising” in the abstract.

2. Recently, articles have been published in the field of optical biosensors using photonic crystals. Considering that Si is also used in these structures, you can also mention them in the text and state the advantage of the proposed work over them. For example:

https://doi.org/10.1364/OL.27.000646, https://doi.org/10.1016/j.optlastec.2021.107397

3. The tables in the text compare previous articles with each other. There is no comparison table of the proposed work with previous articles.

Author Response

Dear Reviewer,

Thank you for taking the time to review this submission, your comments and suggestions are much appreciated.

In response to your comments and suggestions, please find the following:

  1. It is better to revise the text once again and correct typographical errors. For example: “utilising” in the abstract.

Changes were applied to the manuscript.

  1. Recently, articles have been published in the field of optical biosensors using photonic crystals. Considering that Si is also used in these structures, you can also mention them in the text and state the advantage of the proposed work over them. For example: https://doi.org/10.1364/OL.27.000646, https://doi.org/10.1016/j.optlastec.2021.107397

While the topic of photonic crystals is beyond the scope of this paper, we expect to produce future work which will focus on the optics and include this topic within its scope.

  1. The tables in the text compare previous articles with each other. There is no comparison table of the proposed work with previous articles.

This work is highlighting the suitability of SiPMs for Biophotonics applications, the authors are not aware of similar work. For this reason, only a comparison table comparing SiPMs and other similar devices is presented in Table 3.

Kind regards,

Rachel Georgel

Reviewer 3 Report

This manuscript is an overview of Silicon Photomultipliers (SiPMs) with a view to their importance for Bio-photonic and clinical applications, however, the content is superficial or the significance of the content is low, I don’t think it will be interested to community both of SiPMs and Bio-photonics.

 Other drawbacks of this manuscript include:

Line 60-61: “In this section, three main Bio-photonics signal measurements are presented, diffuse 60 reflectance spectroscopy (DRS), Auto-fluorescence (AF) and Raman spectroscopy (RS)”, however, RS is not followed for discussion, or mistake it to AF (native fluorescence).

 Line 138-140: “in order to prevent the device from getting damaged, the avalanche needs to be stopped or limited. This process is known as quenching”. In fact, quenching is necessary not only to prevent the device from getting damaged, but also to restore bias voltage, so that the microcells can detect subsequent photons.

Line 281-312: The authors attributed all the noise of PIN, APD and SiPM to the thermal voltage noise,and make noise performance comparison among them, in fact, due to the fluctuation of avalanche process, the excess noise is significant to APD, and dark count and correlated dark noise are significant to SiPM, those factors should be taken into consideration.

Author Response

Dear Reviewer,

Thank you for taking the time to review this submission, your comments and suggestions are much appreciated.

In response to your comments and suggestions, please find the following:

  1. Line 60-61:“In this section, three main Bio-photonics signal measurements are presented, diffuse 60 reflectance spectroscopy (DRS), Auto-fluorescence (AF) and Raman spectroscopy (RS)”, however, RS is not followed for discussion, or mistake it to AF (native fluorescence).

Currently, unlike DRS and AF, RS is not used for this project but this measurement technique was presented as it is important in Biophotonics. Before the results presented in this paper were found, we did not know the SiPM could be used for RS. However, we have found that the SiPM can be suitable for RS measurement and this will be implemented in future work.

This was now addressed in the manuscript, flagging the possibility to use SiPMs for RS.

  1. Line 138-140: “in order to prevent the device from getting damaged, the avalanche needs to be stopped or limited. This process is known as quenching”. In fact, quenching is necessary not only to prevent the device from getting damaged, but also to restore bias voltage, so that the microcells can detect subsequent photons.

Changes were applied to the manuscript.

  1. Line 281-312: The authors attributed all the noise of PIN, APD and SiPM to the thermal voltage noise, and make noise performance comparison among them, in fact, due to the fluctuation of avalanche process, the excess noise is significant to APD, and dark count and correlated dark noise are significant to SiPM, those factors should be taken into consideration.

In this work, the noise of the different detectors was not compared. The authors compared the thermal noise voltage introduced by the TIA feedback resistance, showing that the SiPM requires smaller resistance values, which, in turn, reduces the amount of noise added to the current to voltage conversion stage. Further clarification and changes were added to section 5 in the manuscript.

Kind regards,

Rachel Georgel

Reviewer 4 Report

The paper is well organized and well written. I have only few concerns which I report here:

1. In the introduction the authors completely neglect the fields of  metasurface for sensing applications. I think that this research area such be mentioned by including some references about silicon membrane for intensity-based sensing.

2. Also, I think that the text in the figures -especially in Figs. 4/5/7 -is too small.

Author Response

Dear Reviewer,

Thank you for taking the time to review this submission, your comments and suggestions are much appreciated.

In response to your comments and suggestions, please find the following:

  1. In the introduction the authors completely neglect the fields of metasurface for sensing applications. I think that this research area such be mentioned by including some references about silicon membrane for intensity-based sensing.

While the topic of metasurfaces is beyond the scope of this paper, we expect to produce future work which will focus on the optics and include this topic within its scope.

  1. Also, I think that the text in the figures -especially in Figs. 4/5/7 -is too small.

Changes were applied to the manuscript, all of the figures were updated with larger text.

Kind regards,

Rachel Georgel

Round 2

Reviewer 3 Report

The merit of this manuscript is an overview on the SiPM in a point of Bio-applications However, there are fatal drawbacks in this manuscript.

1.  Performance comparison of the PIN diode, APD and SiPM is given in Table 3. and Table 4, and PIN and APD are used as important reference. However, as it is well known, limited by the micro cell number, the linear dynamic range of SiPM is much smaller than APD and PIN, the title of this manuscript “Silicon Photomultiplier – A High Dynamic Range, ” is not correct.

2. The authors attributed all the noise of PIN, APD and SiPM to the thermal voltage noise, and make noise performance comparison among them, in fact, due to the fluctuation of avalanche process, the excess noise is significant to APD, and dark count and correlated dark noise are significant to SiPM, the revised manuscript has not taken into consideration.

3. The benefit for SiPM detecting ultra-weak light such as RS should be photon counting method, which has much higher signal to noise ratio than current integration method. However, as mentioned the applications of SiPM in Bio-photonics provided in this manuscript, only current integration method was introduced, it is a biased or misleading message.

Author Response

File Attached.
